# Detection of air and surface contamination by SARS-CoV-2 in hospital rooms of infected patients

Po Ying Chia [1,2,3,11], Kristen Kelli Coleman [4,11], Yian Kim Tan[5,11], Sean Wei Xiang Ong [1,2,11], Marcus Gum[5], Sok Kiang Lau[5], Xiao Fang Lim[5], Ai Sim Lim[5], Stephanie Sutjipto[1,2], Pei Hua Lee[1,2], Than The Son[4], Barnaby Edward Young[1,2,3], Donald K. Milton [6], Gregory C. Gray [4,7,8], Stephan Schuster[9], Timothy Barkham [2,10], Partha Pratim De[2,3], Shawn Vasoo[1,2,3], Monica Chan[1,2], Brenda Sze Peng Ang[1,2,3,10], Boon Huan Tan[5], Yee-Sin Leo[1,2,3,10], Oon-Tek Ng [1,2,3,12✉], Michelle Su Yen Wong[5,12], Kalisvar Marimuthu [1,2,10,12✉] & for the Singapore 2019 Novel Coronavirus Outbreak Research Team*

Understanding the particle size distribution in the air and patterns of environmental contamination of SARS-CoV-2 is essential for infection prevention policies. Here we screen surface and air samples from hospital rooms of COVID-19 patients for SARS-CoV-2 RNA. Environmental sampling is conducted in three airborne infection isolation rooms (AIIRs) in the ICU and 27 AIIRs in the general ward. 245 surface samples are collected. 56.7% of rooms have at least one environmental surface contaminated. High touch surface contamination is shown in ten (66.7%) out of 15 patients in the first week of illness, and three (20%) beyond the first week of illness ($p = 0.01$, $\chi^2$ test). Air sampling is performed in three of the 27 AIIRs in the general ward, and detects SARS-CoV-2 PCR-positive particles of sizes >4 μm and 1–4 μm in two rooms, despite these rooms having 12 air changes per hour. This warrants further study of the airborne transmission potential of SARS-CoV-2.

[1] National Centre for Infectious Diseases, Singapore, Singapore. [2] Tan Tock Seng Hospital, Singapore, Singapore. [3] Lee Kong Chian School of Medicine, Nanyang Technological University, Singapore, Singapore. [4] Duke-NUS Medical School, National University of Singapore, Singapore, Singapore. [5] DSO National Laboratories, Singapore, Singapore. [6] Maryland Institute for Applied Environmental Health, University of Maryland School of Public Health, College Park, MD, USA. [7] School of Medicine and Global Health Institute, Duke University, Durham, NC, USA. [8] Global Health Research Center, Duke Kunshan University, Kunshan, China. [9] Singapore Centre for Environmental Life Sciences Engineering, Nanyang Technological University, Singapore, Singapore. [10] Yong Loo Lin School of Medicine, National University of Singapore, Singapore, Singapore. [11]These authors contributed equally: Po Ying Chia, Kristen Kelli Coleman, Yian Kim Tan, Sean Wei Xiang Ong. [12]These authors jointly supervised this work: Oon-Tek Ng, Michelle Su Yen Wong, Kalisvar Marimuthu. *A list of authors and their affiliations appear at the end of the paper. ✉email: Oon_Tek_Ng@ncid.sg; kalisvar_marimuthu@ncid.sg

Severe acute respiratory syndrome coronavirus 2 (SARS-CoV-2) causing coronavirus disease 2019 (COVID-19) has spread globally and many countries are experiencing ongoing local transmission despite varying levels of control efforts. Understanding the different transmission routes of SARS-CoV-2 is crucial in planning effective interventions to break the chain of transmission. Although extensive surface contamination with SARS-CoV-2 by a symptomatic patient has been demonstrated[1], little is known about airborne transmission of SARS-CoV-2. It is also unknown if asymptomatic individuals pose the same environmental contamination risk as symptomatic ones, although viral shedding has been demonstrated to continue even after clinical recovery of COVID-19 patients[2]. There are multiple reports of asymptomatic patients testing positive for SARS-CoV-2[3,4], and the potential transmission of the virus by an asymptomatic person has been described[5]. Therefore, viral contamination of the air and surfaces surrounding asymptomatic or recovering COVID-19 patients could have serious implications for outbreak control strategies. This knowledge gap is recognized in the Report of the WHO-China Joint Mission on Coronavirus 2019[6].

The primary objective of our study is to identify potential patient-level risk factors for environmental contamination by SARS-CoV-2 by sampling the air and surfaces surrounding hospitalized COVID-19 patients at different stages of illness.

## Results

**Air and environmental sampling.** Environmental sampling was conducted in three airborne infection isolation rooms (AIIRs) in the ICU and 27 AIIRs in the general ward. Air sampling was performed in three of the 27 AIIRs in the general ward. All patients reported COVID-19 symptoms. Seven patients (23%) were asymptomatic at the time of environmental sampling. Of the 23 symptomatic patients, 18 (78%) had respiratory symptoms, 1 had gastrointestinal symptoms, 1 had both respiratory and gastrointestinal symptoms, and 3 patients (10%) had fever or myalgia only (Supplementary Table 1).

Air samples from two (66.7%) of three AIIRs tested positive for SARS-CoV-2, in particle sizes >4 μm and 1–4 μm in diameter (Table 1). Samples from the fractionated size <1 μm were all negative, as were all non-size-fractionated SKC polytetrafluoroethylene (PTFE) filter cassette samples. Total SARS-CoV-2 concentrations in air ranged from $1.84 \times 10^3$ to $3.38 \times 10^3$ RNA copies per $m^3$ air sampled. Rooms with viral particles detected in the air also had surface contamination detected.

There were no baseline differences between patients with environmental surface contamination and those without, in terms of age, comorbidities, and positive clinical sample on the day of sampling. Median cycle threshold (Ct) values of the clinical specimens for patients with and without environmental surface contamination were 25.69 (IQR 20.37–34.48) and 33.04 (28.45–35.66), respectively (Table 2).

Of the rooms with environmental contamination, the floor was most likely to be contaminated (65%), followed by the air exhaust vent (60%, $n = 5$), bed rail (59%), and bedside locker (47%) (Fig. 1). Contamination of toilet seat and automatic toilet flush button was detected in 5 out of 27 rooms, and all 5 occupants had reported gastrointestinal symptoms within the preceding 1 week of sampling. We did not detect surface contamination in any of the three ICU rooms.

Presence of environmental surface contamination was higher in week 1 of illness (Fig. 2) and showed association with the clinical cyclical threshold ($P = 0.06$, Wilcoxon rank-sum test). Surface environment contamination was not associated with the presence of symptoms (Table 2). In a subgroup analysis, the presence and extent of high-touch surface contamination were significantly higher in rooms of patients in their first week of illness (Fig. 2). The best fit curve with the least-squares fit (Fig. 3) showed that the extent of high-touch surface contamination declined with increasing duration of illness and Ct values. There was also no correlation between the Ct values of clinical samples and the Ct values of environmental samples across the days of illness (Supplementary Fig. 3).

## Discussion

Surface sampling revealed that the PCR-positivity high-touch surfaces was associated with nasopharyngeal viral loads and peaked at approximately day 4–5 of symptoms. Air sampling of the AIIR environments of two COVID-19 patients (both day 5 of illness with high nasopharyngeal swab viral loads) detected the presence of SARS-CoV-2 particles sized 1–4 μm and >4 μm. The absence of any detection of SARS-CoV-2 in air samples of the third patient (day 9 of illness with lower nasopharyngeal viral load concentration) suggests that the presence of SARS-CoV-2 in the air is possibly highest in the first week of illness.

Recent aggregated environmental sampling and laboratory experiments have examined the particle size distribution of

**Table 1 Severe acute respiratory syndrome coronavirus 2 (SARS-CoV-2) detections in the air of hospital rooms of infected patient.**

| Patient | Day of illness | Symptoms reported on day of air sampling | Clinical Ct value[a] | Airborne SARS-CoV-2 concentrations (RNA copies $m^{-3}$ air) | Aerosol particle size | Samplers used |
|---------|----------------|------------------------------------------|----------------------|---------------------------------------------------------------|-----------------------|---------------|
| 1 | 9 | Cough, nausea, dyspnea | 33.22 | ND | >4 μm | NIOSH |
|   |   |                        |       | ND | 1–4 μm | |
|   |   |                        |       | ND | <1 μm | |
|   |   |                        |       | ND | – | SKC filters |
| 2 | 5 | Cough, dyspnea | 18.45 | 2,000 | >4 μm | NIOSH |
|   |   |                |       | 1,384 | 1–4 μm | |
|   |   |                |       | ND | <1 μm | |
| 3 | 5 | Asymptomatic[b] | 20.11 | 927 | >4 μm | NIOSH |
|   |   |                 |       | 916 | 1–4 μm | |
|   |   |                 |       | ND | <1 μm | |

ND none detected.
[a]PCR cycle threshold value from patient's clinical sample.
[b]Patient reported fever, cough, and sore throat until the day before the sampling. Patient reported no symptoms on the day of sampling, however was observed to be coughing during sampling.

**Table 2 Baseline clinical characteristics of COVID-19 patients with environmental contamination.**

| Characteristics of COVID-19 patients | Rooms with surface environment contamination (n = 17) | Rooms without surface environment contamination (n = 13) | P value |
|---|---|---|---|
| Median age (IQR) | 52 (42–62) | 44 (36–55) | 0.75 |
| Male Sex (%) | 6 (46%) | 8 (47%) | 0.96 |
| Median Age Adjusted Charlson's Comorbidity Index (IQR) | 1 (0–2) | 1 (0–1) | 0.69 |
| Median day of Illness (IQR) | 5 (4–9) | 13 (5–20) | 0.17 |
| Median day of stay in room (IQR) | 3 (3–8) | 4 (2–16) | 0.95 |
| Oxygen requirement (%) | 0 | 4 (31) | 0.03 |
| Symptomatic (%) | 12 (71) | 11 (85) | 0.43 |
| Respiratory symptoms (%) | 11 (65) | 7 (54) | 0.55 |
| Gastrointestinal symptoms (%) | 1 (6) | 1 (8) | >0.99 |
| Clinical Cycle threshold value, median (IQR)[a] | 25.69 (20.37–34.48) | 33.04 (28.45–35.66) | 0.06 |

[a]PCR cycle threshold value from patient's clinical sample.
$\chi^2$ or Fisher's exact test was used to compare categorial variables; and Student's t test or nonparametric Wilcoxon rank-sum was used to compare continuous variables.

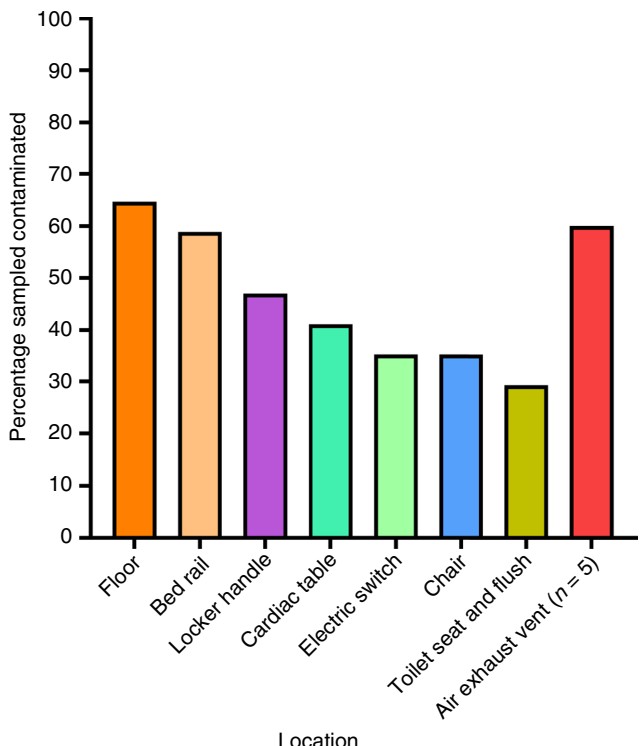

**Fig. 1 Percentage of contaminated swabs from surface samples, in rooms with any contamination.** All sites were n = 17, except for air exhaust vents where n = 5.

**SARS-CoV-2 in the air.** A study from Wuhan, China sampled three different environmental settings and detected aerosol size range particles.[7] Additionally, a recent laboratory study demonstrated the ability of SARS-CoV-2 to remain viable in aerosols for up to 3 h.[8] Although limited in subject numbers, our study examined this issue at the individual patient-level, thus enabling correlation of particle size distribution in the air with symptoms duration and nasopharyngeal viral loads. The absence of aerosol-generating procedures or intranasal oxygen supplementation reduces the possibility of our current findings being iatrogenic in nature. Larger individual patient-level studies examining the droplet and aerosolizing potential of SARS-CoV-2 over different distances and under different patient and environmental conditions are rapidly needed to determine the generalizability of our current findings.

Contrary to the study from Wuhan, China that detected SARS-CoV-2 in aerosols 0.25–1.0 μm in diameter[7], the smallest aerodynamic size fraction that contained detectable levels of SARS-CoV-2 in our study was 1–4 μm. The non-detection of SARS-CoV-2 in particles <1 μm could have been due to the reduced efficiency of extracting viruses from filters as compared with extracting viruses adhered to the wall of the 1.5 mL and 15 mL centrifuge tubes, where particles 1–4 μm and >4 μm in diameter are captured using the NIOSH aerosol sampler. Furthermore, to our knowledge, this is the first time the NIOSH samplers have been used to capture coronaviruses. Therefore, no baseline data exist for airborne coronavirus sampling using these samplers, limiting our understanding of the negative results in the <1 μm size fraction.

The extent of environmental contamination we found in our study could be attributable to direct touch contamination by either the patient or healthcare workers after contact with infected respiratory fluids. However, contamination through respiratory droplets emitted through coughing and sneezing, as well as through respiratory aerosols, is also plausible. Contamination of surface sites not frequently touched (air exhaust vents and floor) support this latter hypothesis.

In the current analysis, the presence and concentration of SARS-CoV-2 in air and high-touch surface samples correlated with the day of illness and nasopharyngeal viral loads of COVID-19 patients. This finding is supported by multiple observational clinical studies, which have demonstrated that SARS-CoV-2 viral loads peak in the first week among COVID-19 patients[2,9,10], with active viral replication in the upper respiratory tract in the first 5 days of illness[11]. This finding could help inform public health and infection prevention measures in prioritizing resources by risk stratifying COVID-19 patients by their potential to directly or indirectly transmit the SARS-CoV-2 virus to others.

Our study was limited in that it did not determine the ability of SARS-CoV-2 to be cultured from the environmental swabs and the differentially sized air particles, which would be vital to determining the infectiousness of the detected particles. Another study from Nebraska attempted virus culture on SARS-CoV-2 PCR-positive air samples, however could not isolate viable virus[12]. The difficulty in culturing virus from air samples arises from low-virus concentrations, as well as the compromised integrity of the virus due to air sampling stressors. Future studies using enhanced virus culture techniques could be considered[13], and efforts to design a culture method to isolate virus from our

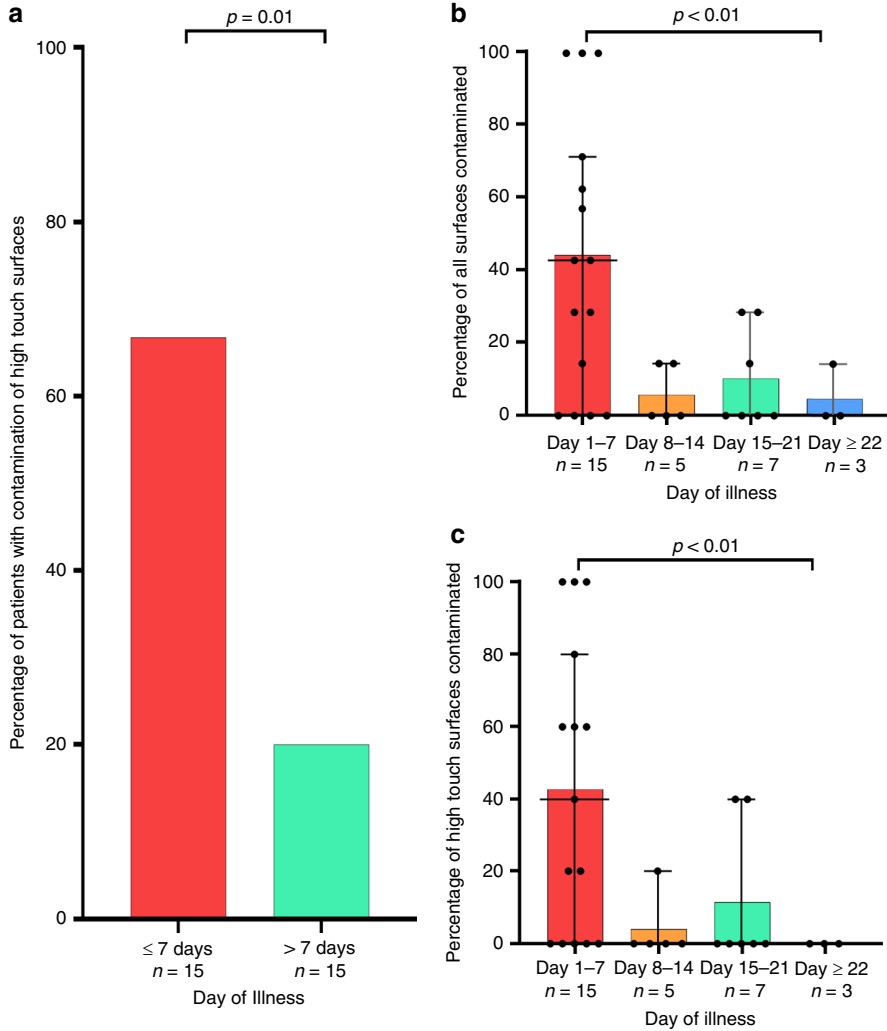

**Fig. 2 Extent of environmental contamination correlated with day of illness timepoint. a** Percentage of patients with contamination of high-touch surfaces in the first week of illness compared with more than first week of illness, $n = 15$ in both groups. **b** Percentage of surfaces contaminated across weeks of illness with median and 95% confidence intervals. **c**. Percentage of high-touch surfaces contaminated across weeks of illness with median and 95% confidence intervals.

samples is underway. Second, sampling in an AIIR environment may not be representative of community settings and further work is needed to generalize our current findings. Third, we sampled each room at a single timepoint during the course of illness and did not track environmental contamination over the course of illness for individual patients. Fourth, as clinical results were within 72 h of environmental testing, it is plausible that during the day of testing, viral load was actually low or negligible, hence limiting environmental contamination.

Current evidence does not seem to point to aerosolization as the key route of transmission of SARS-CoV-2, and there have been reports of healthcare workers not being infected after exposure to confirmed patients despite not using airborne precautions[14]. Detailed epidemiologic studies of outbreaks, in both healthcare and non-healthcare settings, should be carried out to determine the relative contribution of various routes of transmission and their correlation with patient-level factors.

In conclusion, in a limited number of AIIR environments, our current study involving individual COVID-19 patients not undergoing aerosol-generating procedures suggests that SARS-CoV-2 can be shed in the air from a patient in particles sized between 1 and 4 microns. Even though particles in this size range have the potential to linger longer in the air, more data on

viability and infectiousness of the virus would be required to confirm the potential airborne spread of SARS-CoV-2. Additionally, the concentrations of SARS-CoV-2 in the air and high-touch surfaces could be highest during the first week of COVID-19 illness. Further work is urgently needed to examine these findings in larger numbers and different settings to better understand the factors affecting air and surface spread of SARS-CoV-2 and inform effective infection prevention policies.

## Methods

**Study design, patient selection, and data collection**. We conducted this cross-sectional study in AIIRs at the National Centre for Infectious Diseases, Singapore. These rooms had 12 air changes per hour, an average temperature of 23 °C, relative humidity of 53–59%, and exhaust flow of 579.6 m³/h.

Patients with a SARS-CoV-2 infection confirmed by a polymerase chain reaction (PCR)-positive respiratory sample within the prior 72 h were included. Clinical characteristics, including the presence of symptoms, day of illness, day of stay in the room, supplemental oxygen requirement, and baseline characteristics, were collected. One patient from a previously published pilot study on environmental sampling in the same facility (Patient 30; Supplementary Table 1) was also included in the current analysis[1].

**Air sampling**. Six NIOSH BC 251 bioaerosol samplers were placed in each of three AIIRs in the general ward to collect air samples (set-up depicted in Supplementary Fig. 1). Particles collected with the NIOSH sampler are distributed into three size

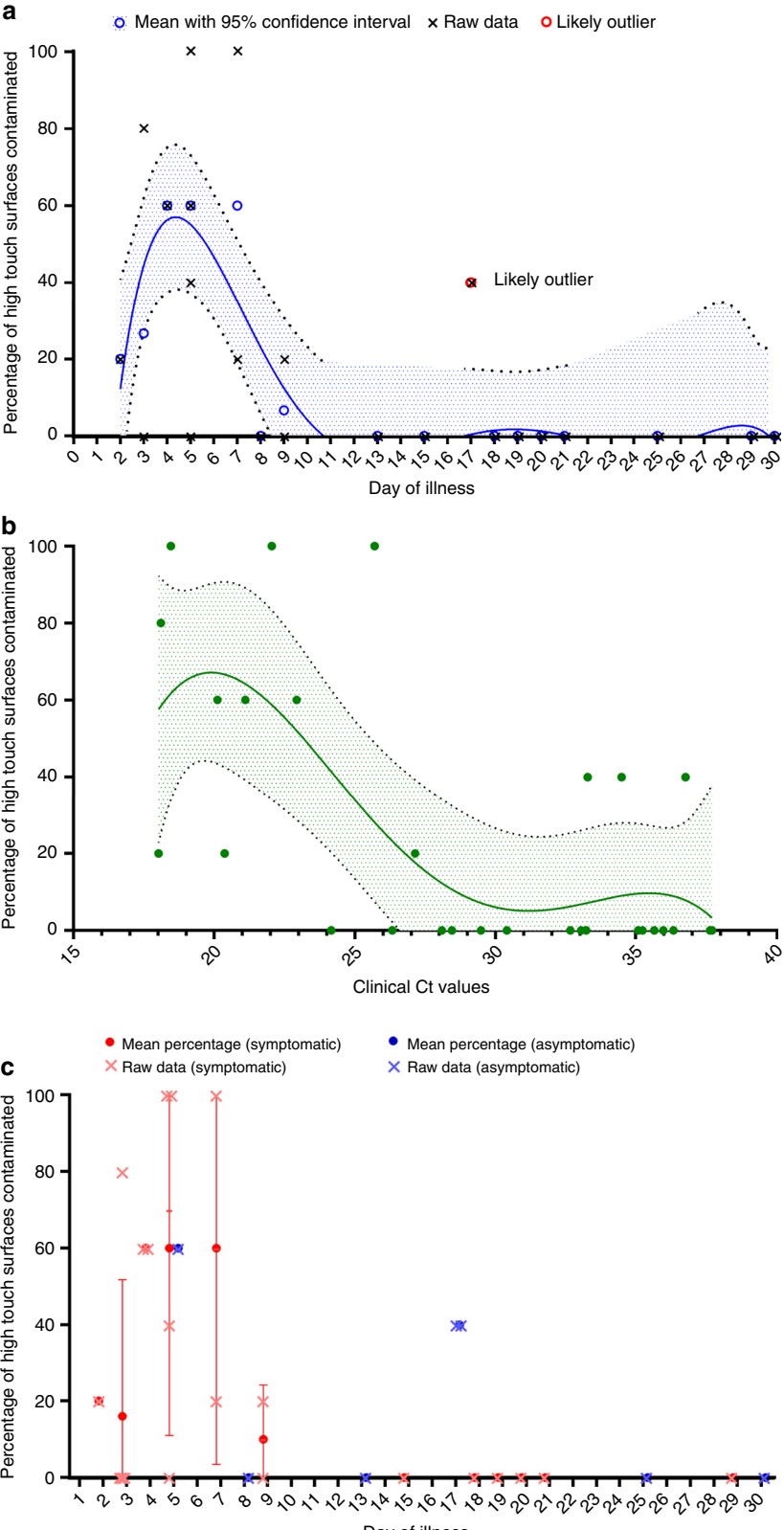

**Fig. 3 Patient and disease factors affecting percentage of high-touch contamination. a** Mean percentage of high-touch surface contaminated by day of illness with 95% confidence interval with best fit curve, $n = 30$. **b** Percentage of high-touch surfaces contaminated by clinical cycle threshold values with 95% confidence interval with bestfit curve, $n = 30$. **c** Mean percentage of high-touch surface contaminated by day of illness with 95% confidence interval grouped by symptoms, $n = 30$.

fractions. Particles >4 μm in diameter are collected in a 15 mL centrifuge tube, particles 1–4 μm in diameter are collected in a 1.5 mL centrifuge tube, and particles <1 μm in diameter are collected in a self-assembled filter cassette containing a 37-mm diameter, PTFE filter with 3 μm pores. All NIOSH samplers were connected to either SKC AirCheck TOUCH Pumps or SKC Universal air sampling pumps set at a flow-rate of 3.5 L/min and run for 4 h, collecting a total of 5040 L of air from each patient's room.

In the room of Patient 1, three NIOSH samplers were attached to each of two tripod stands and situated at different heights from the ground (1.2 m, 0.9 m, and 0.7 m) near the air exhaust to capture particles from the unidirectional airflow in the room. Throughout the 4-hour sampling period, Patient 1 was intermittently facing the NIOSH samplers, whereas seated 1 meter from the first tripod and 2.1 meters from the second tripod. Four SKC 37 mm PTFE filter (0.3 μm pore size) cassettes were also distributed throughout the room and connected to SKC Universal air sampling pumps set at a flow-rate of 5 L/min, each collecting an additional 1200 L of air from the room.

In the rooms of Patients 2 and 3, three NIOSH samplers were attached to each of two tripod stands and situated at different heights from the ground (1.2 m, 0.9 m, and 0.7 m). Throughout the 4-hour sampling period, Patients 2 and 3 remained in bed within 1 meter from all six NIOSH samplers (Supplementary Fig. 1). Patient 3 was also talking on the phone for a significant proportion of time during sampling. Additional SKC pumps with PTFE filter cassettes were not used in the rooms of Patient 2 and 3.

The six NIOSH samples from each room were pooled prior to analysis, but the particle size fractions remained separated. Each sample pool was representative of 5040 L air.

**Surface sampling**. Surface samples were collected with Puritan EnviroMax Plus pre-moistened macrofoam sterile swabs (25-88060). Eight to 20 surface samples were collected from each room. Five surfaces were designated high-touch surfaces, including the cardiac table, entire length of the bed rails including bed control panel and call bell, bedside locker, electrical switches on top of the beds, and chair in general ward rooms (Supplementary Fig. 1). In ICU rooms, the ventilator and infusion pumps were sampled instead of the electrical switches on top of the beds and chair (Supplementary Fig. 2). Air exhaust outlets and glass window surfaces were sampled in five rooms, including the three rooms in which air sampling was performed. Toilet seat and automatic flush button (one combined swab) were sampled in AIIR rooms in the general ward. Supplementary Table 2 lists all surface samples.

**Sample transfer and processing**. All samples were immediately stored at 4 °C in the hospital prior to transfer to a BSL-3 laboratory where samples were immediately processed and stored at −80 °C unless directly analyzed. Prior to RNA extraction, NIOSH aerosol sample tubes and filters were processed as previously described[15], with slight modification due to the pooling of samples.

**Laboratory methods**. The QIAamp viral RNA mini kit (Qiagen Hilden, Germany) was used for sample RNA extraction. Real-time PCR assays targeting the envelope (E) genes[16] and an orf1ab assay modified from Drosten et al.[17] were used to detect SARS-CoV-2 in the samples[18]. In brief, for the envelope gene assay, a 20 μl reaction mix was prepared with 12.5 μl of SuperScript III Platinum One-Step qRT-PCR Kit (Thermofisher Scientific, USA) buffer, 0.75 mM $Mg_2SO_4$, 5 μl of RNA, 400 nM each of the forward primer (E_Sarbeco_F1- ACAGGTACGTTAATAGTTAATAG CGT) and reverse primer (E_Sarbeco_R2- ATATTGCAGCAGTACGCACACA) with 200 nM of probe (E_Sarbeco_P1- (FAM) ACACTAGCCATCCTTACTGCGC TTCG (BHQ1)). Thermal cycling conditions included reverse transcription at 55 °C for 10 min, an initial denaturation at 95 °C for 5 min, followed by 45 cyles of 95 °C for 15 s, 58 °C for 1 min. For the orf1ab assay, a 20 μl reaction mix was prepared with 12.5 μl of SuperScript III Platinum One-Step qRT-PCR Kit (Thermofisher Scientific, USA) buffer, 0.5 mM $Mg_2SO_4$, 5 μl of RNA, 800 nM each of the forward primer (Wu-BNI-F- CTAACATGTTTATCACCCGCG) and reverse primer (Wu-BNI-R- CTCTAGTAGCATGACACCCCTC) with 400 nM of probe (WU-BNI-P- (FAM) TAAGACATGTACGTGCATGGATTGGCTT (BHQ1)). Thermal cycling conditions included reverse transcription at 55 °C for 10 min, an initial denaturation at 95 °C for 5 min, followed by 45 cyles of 95 °C for 15 s, 60 °C for 1 min. All samples were run in duplicate and with both assays. Positive detection was recorded as long as amplification was observed in at least one assay.

**Cleaning regimen of rooms**. Routine environmental cleaning of the rooms was carried out by a trained team of housekeeping staff. High-touch surfaces (e.g., bed rail, cardiac table, switches) were cleaned twice daily using 5000 parts per million (ppm) sodium dichloroisocyanurate (NaDCC), reconstituted using Biospot Effervescent Chlorine Tablets. The floor was cleaned daily using 1000 ppm NaDCC. All surface sampling was performed in the morning before the first cleaning cycle for the day.

**Statistical analysis**. Statistical analysis was performed using Stata version 15.1 (StataCorp, College Station, Texas) and GraphPad Prism 8.0 (GraphPad Software, Inc., San Diego). $P < 0.05$ was considered statistically significant, and all tests were

two-tailed. For the surface environment, outcome measures analyzed were any positivity by room and pooled percentage positivity by day of illness and respiratory viral load (represented by clinical cycle threshold (Ct) value). We analyzed the factors associated with environmental contamination using the Student $t$ test, or the nonparametric Wilcoxon rank-sum test was used for continuous variables depending on their distribution. The $\chi^2$ or Fisher exact test was used to compare categorical variables. We plotted the best fit curve by least-square method to study the environmental contamination distribution across various the days of illness and clinical Ct value.

**Ethics statement**. Informed consent was waived as clinical data were collected as part of outbreak investigation under the Infectious Diseases Act, authorized by the Ministry of Health, Singapore. The clinical data were collected by a study team member who was appointed by the Ministry of Health, Singapore as a Public Health Officer and authorized for the collection of anonymized clinical data as part of the outbreak investigation. All clinical data were collected using a standardized anonymized structured case report form with no patient identifiers recorded, and stored on a secured server.

**Reporting summary**. Further information on research design is available in the Nature Research Reporting Summary linked to this article.

## Data availability

The data sets generated during and/or analyzed during the current study are available from the corresponding author on reasonable request. The source data underlying Figs. 1, 2a–c, 3a–c, and Supplementary Fig. 3 are provided as Source Data file. Source data are provided with this paper.

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

## Acknowledgements

We thank Dr. Bill Lindsley (National Institute for Occupational Safety and Health, NIOSH) for loaning us aerosol samplers and guiding us in their use; Dr. Raquel Binder and Emily Robie from the Duke One Health team for quickly providing us equipment for use in this study; Dr Hsu Li Yang (Saw Swee Hock School of Public Health, National University of Singapore) for facilitating the air sampling; and Dr Ding Ying (National Centre for Infectious Diseases, Singapore) for helping with project coordination. No compensation was received for their roles in the study. We further thank the colleagues in DSO National Laboratories in the environmental detection team and clinical diagnostics team for BSL-3 sample processing and analysis as well as the logistics and repository team for transport of biohazard material, inventory, and safekeeping of received items. This study is funded by NMRC Seed Funding Program (TR19NMR119SD), NMRC COVID-19 Research Fund (COVID19RF-001, COVID19RF-002), NHG-NCID COVID-19 Centre Grant (COVID19 CG0002), and internal funds from DSO National Laboratories. Ng O.T. is supported by NMRC Clinician Scientist Award (MOH-000276). K. Marimuthu is supported by NMRC CS-IRG (CIRG18Nov-0034). Chia P.Y. is supported by NMRC Research Training Fellowship (NMRC/Fellowship/0056/2018).

## Author contributions

K.K.C., Y.K.T., S.W.X.O., O.-T.N., M.S.Y.W., and K.M. designed the study. P.Y.C., S.W.X.O., St.Su., P.H.L., O.-T.N., and K.M. collected the air and environmental samples. K.K.C., T.T.S., and G.C.G. provided the technical support and equipment for air sampling. B.E.Y. collected the clinical data. P.Y.C. conducted the data analysis. Y.K.T., M.G., S.K.L., X.F.L., A.S.L., T.B., P.P.D., B.H.T., and M.S.Y.W. conducted the laboratory testing. D.K.M., G.C.G., St.Sc., and B.H.T. provided technical guidance and advice. G.C.G., S.V., M.C., B.S.P.A., B.H.T., and Y.-S.L. provided overall supervision. P.Y.C., K.K.C., S.W.X.O., M.S.Y.W., and K.M. drafted the manuscript. All authors read and approved the final manuscript.

## Competing interests

The authors have no competing interests as defined by Nature Research, or other interests that might be perceived to influence the results and/or discussion reported in this paper. The funders had no role in the design and conduct of the study; collection, management, analysis, and interpretation of the data; preparation, review, or approval of the manuscript; and decision to submit the manuscript for publication.

## Additional information

## for the Singapore 2019 Novel Coronavirus Outbreak Research Team

David Chien Lye[1,2,3,10], Poh Lian Lim[1,2,3,10], Cheng Chuan Lee[1,2], Li Min Ling[1,2], Lawrence Lee[1,2], Tau Hong Lee[1,2], Chen Seong Wong[1,2], Sapna Sadarangani[1,2], Ray Junhao Lin[1,2], Deborah Hee Ling Ng[1,2], Mucheli Sadasiv[1,2], Tsin Wen Yeo[1,2], Chiaw Yee Choy[1,2], Glorijoy Shi En Tan[1,2], Frederico Dimatatac[1,2], Isais Florante Santos[1,2], Chi Jong Go[1,2], Yu Kit Chan[1,2], Jun Yang Tay[1,2], Jackie Yu-Ling Tan[2], Nihar Pandit[2], Benjamin Choon Heng Ho[2], Shehara Mendis[2], Yuan Yi Constance Chen[2], Mohammad Yazid Abdad[1] & Daniela Moses[9]

