## [Peer Review File · Nature Communications]

Reviewers' Comments:

Reviewer #1:

Remarks to the Author:

This work experimentally characterized SARS-CoV-2 on several types of environmental surfaces and in a smaller number of air samples deployed in airborne infection isolation rooms (AIIRs) of COVID-19 patients in a hospital in Singapore. Surface sampling was conducted in 3 AIIRs in the ICU and 27 AIIRs in the general ward, and air sampling was conducted in 3 of the 27 AIIRs in the general ward. Air sampling was conducted using NIOSH bioaerosol samplers that collect 1-4 μm and $>4 \mu\text{m}$ particles into liquid in centrifuge, and $<1 \mu\text{m}$ particles onto PTFE filters (although I never saw the $<1 \mu\text{m}$ results). Some additional non-size-segregated air sampling was also done using PTFE filter cassettes and pumps (although I never saw those results). Surface sampling was conducted using pre-moistened foam sterile swabs. Key novel findings include: 2 of the 3 rooms with air samples had SARS-CoV-2 PCR-positive results in the $>4 \mu\text{m}$ and 1-4 μm size ranges (even in an environment with very high air change rate of 12 per hour); over half the rooms had at least one environmental surface test positive; high touch surface contamination was higher during the first week of illness than beyond the first week of illness; and that floor contamination was most common followed by bed rail. For the most part these are novel and useful findings in this current pandemic. I have several suggestions for improving the paper below.

Line by line comments:

Line 38: "in 3 rooms" – suggesting be clear that these were a subset of 3 of the 30 airborne infection isolate rooms and not 3 "other rooms"

Line 39-40 (abstract): I think that in the abstract results portion it should be mentioned that air sampling detected PCR-positive particles in two rooms, "despite the airborne infection isolation rooms having 12 air changes per hour" or something to that effect. That to me is a key finding – that viral RNA was found in air samples (and in both size fractions) even at extremely high air change routes found basically only in AIIRs. That means their findings would be expected to be magnified in more typical indoor environments (as the authors mention later on line 199-200).

Line 80: since the $<1 \mu\text{m}$ size fraction collects particles onto different media (solid PTFE rather than into centrifuge with liquid), there could be differences in extraction efficiencies between the $<1 \mu\text{m}$ fractions and $>1 \mu\text{m}$ fractions that contribute to differences in magnitude of RNA copies recovered. Do the authors have any insight into the potential for differences in extraction efficiencies between these two collection types and how they might manifest in the results? My concern is that if only a small fraction of viral RNA can be recovered from the PTFE filters but nearly 100% is recovered from the centrifuges, then the true magnitude of RNA copies in the small size fraction would be inadvertently masked and lead to misleading outcomes in that size fraction.

Line 84-88: should refer to the SI directly here because it is difficult to interpret the sampling setup without seeing Figure S1 (or it seems that there is no Figure in the SI that corresponds to Patient 1 but only patients 2 and 3? This is confusing to me).

Figure s1: Also in Figure S1 it is unclear what "A" means in red circles. "A" = "air sampler" perhaps? Please clarify in the caption.

Line 88-91: I didn't see the location of these other non-size-segregated SKC 37 mm PTFE filter cassettes in the SI diagram of sampling. Please add and generally improve the visual description of the sampling campaign to help best interpret findings. (I also never saw results from these samples

presented anywhere either).

Line 98: pooling of the 6 NIOSH samples... why was this done? It seems a missed opportunity to get some vertical variations. Were you worried about recovering enough viral copies to detect?

Line 101-102: what area of surface was swabbed and how does it compare to typical guidance? For example, WHO recommends sampling 25 cm² (https://apps.who.int/iris/bitstream/handle/10665/331058/WHO-2019-nCoV-Environment_protocol-2020.1-eng.pdf?sequence=1&isAllowed=y). Was at least a consistent amount of area swabbed? This could have important implications for the finding in Figure S3 that clinical and environmental Ct values were not correlated over time (i.e., if different surfaces were swabbed with different areas then that could be masking some association, which would just mean that clear conclusions can't necessarily be drawn).

Line 134-135: this specific description of environmental sampling in 3 AIIRs in the ICU and 27 AIIRs in the general ward, and also air sampling in 3 of the 27 AIIRs in the general ward, is very useful language and I suggest it be used in the abstract to increase specificity in abstract readers of the description of where samples were taken and where virus was recovered.

Lines 141-142: this is where concentration estimates could vary potentially if viral recovery from the <1µm filter was low, although I was surprised to not see any data reported from the <1 µm size range in Table 1. These results are surprisingly just not shown, which is a big omission in my opinion. Why weren't they shown?

Line 149-153 and Figure 1: no quantification of concentrations was conducted for environmental contamination? (i.e., copies per cm²) Why not?

Reviewer #2:

Remarks to the Author:

This paper includes important information on virus RNA detections in the air and surfaces in patient rooms in Singapore. The sample size of 30 patients is larger than some previous studies, but a greater number of measurements particularly longitudinal measurements would have strengthened the study. There are a number of limitations to the conclusions

1. Virus was not cultured and therefore infectivity cannot be confirmed. While air samples are known to be difficult to culture (as noted in lines 195-7) that would not apply to the surface samples?

2. Was the greatest frequency of surface contamination actually on the non-touch surfaces (Figure 1), suggesting that even on the high-touch surfaces the presence of virus could have been through deposition of droplets rather than by contaminated touches? The lack of surface contamination in the highly-ventilated ICU rooms further supports this hypothesis?

3. Figure 3 would be easier to interpret if longitudinal data were available.

Reviewer #3:

Remarks to the Author:

In this manuscript the authors describe a study of 30 AIIR rooms occupied by COVID-19 patients and

tested for SARS-CoV-2 to identify risk factors for environmental contamination by the virus. 245 surface samples from 27 rooms and aerosols samples in 3 rooms were examined showing 57% of rooms having at least 1 positive environmental sample and 2 of 3 rooms with positive airborne samples with particle in the 1-4 and >4 um size range. The data shows surface contamination is more prevalent in the first week of symptoms and the positive aerosol samples corroborates potential airborne transmission of the disease. The conclusions while not novel (multiple papers have been published) provides important convincing evidence and data that is needed by the HealthCare community in response to this rapidly moving respiratory disease.

Comments:

- 1) The paper only presents PCR data and would be greatly enhanced by attempts to culture the virus to show infectivity.
- 2) More description of the RT-PCR methods is needed, RT-PCR targeted envelope genes and in house orf lab assay were used. Are they validated assays, how was the q-PCR done, e.g. the aerosol data is presented as the number of RNA copies/m³ with no information regarding what reference material is used for the qPCR.
- 3) While the statistics seem appropriate, the description is limited. Figure 1 and 2 do not have an indication for variability only significance.
- 4) For the surface samples some indication of the amount of area sampled should be provided. Was a standard template used or just random swabbing of the surface.
- 5) The data in table 2 suggests an inverse relationship between environmental contamination and O₂ use, yet in line 156 the authors claim environmental contamination was not associated with the presence of symptoms or supplementary oxygen and in line 213 use the lack of association to conclude that in spite of O₂ supplementation patients can still shed virus aerosols. This discrepancy should be cleared up.
- 6) Line 149-150 should indicate that the air exhaust vent was also a location of significant contamination.
- 7) I think the fact that many of the contact surfaces that have been tested are in close proximity to the patients and could be contaminated by large droplets from patient coughs and sneezes and are not necessarily contaminated by touch needs to be mentioned in the discussion, the data could be demonstrating droplet spray and aerosol spread but the emphasis on "high touch" surfaces detracts from this concept.
- 8) minor typos. line 308 in in and line 315 95% not 955

REVIEWER #1:

This work experimentally characterized SARS-CoV-2 on several types of environmental surfaces and in a smaller number of air samples deployed in airborne infection isolation rooms (AIIRs) of COVID-19 patients in a hospital in Singapore. Surface sampling was conducted in 3 AIIRs in the ICU and 27 AIIRs in the general ward, and air sampling was conducted in 3 of the 27 AIIRs in the general ward. Air sampling was conducted using NIOSH bioaerosol samplers that collect 1-4 μm and $>4 \mu\text{m}$ particles into liquid in centrifuge, and $<1 \mu\text{m}$ particles onto PTFE filters (although I never saw the $<1 \mu\text{m}$ results). Some additional non-size-segregated air sampling was also done using PTFE filter cassettes and pumps (although I never saw those results). Surface sampling was conducted using pre-moistened foam sterile swabs. Key novel findings include: 2 of the 3 rooms with air samples had SARS-CoV-2 PCR-positive results in the $>4 \mu\text{m}$ and 1-4 μm size ranges (even in an environment with very high air change rate of 12 per hour); over half the rooms had at least one environmental surface test positive; high touch surface contamination was higher during the first week of illness than beyond the first week of illness; and that floor contamination was most common followed by bed rail. For the most part these are novel and useful findings in this current pandemic. I have several suggestions for improving the paper below.

Reply: Thank you for these helpful suggestions. The $<1 \mu\text{m}$ particles, as well as the non-size-segregated PTFE filter cassettes samples, were all negative. This has been included in the results section (lines 150-151) and is now more clearly outlined in Table 1.

Line 38: “in 3 rooms” – suggesting be clear that these were a subset of 3 of the 30 airborne infection isolate rooms and not 3 “other rooms”

Reply: Clarification has been made that these 3 rooms were a subset of the 27 general ward rooms (line 43).

Line 39-40 (abstract): I think that in the abstract results portion it should be mentioned that air sampling detected PCR-positive particles in two rooms, “despite the airborne infection isolation rooms having 12 air changes per hour” or something to that effect. That to me is a key finding – that viral RNA was found in air samples (and in both size fractions) even at extremely high air change routes found basically only in AIIRs. That means their findings would be expected to be magnified in more typical indoor environments (as the authors mention later on line 199-200).

Reply: Thank you for this suggestion. This additional statement has been added to the abstract (line 45).

Line 80: since the $<1 \mu\text{m}$ size fraction collects particles onto different media (solid PTFE rather than into centrifuge with liquid), there could be differences in extraction efficiencies between the $<1 \mu\text{m}$ fractions and $>1 \mu\text{m}$ fractions that contribute to differences in magnitude of RNA copies recovered. Do the authors have any insight into the potential for differences in extraction efficiencies between these two collection types and how they might manifest in the results? My concern is that if only a small fraction of viral RNA can be recovered from the PTFE filters but nearly 100% is recovered from the centrifuges, then the true magnitude of RNA copies in the small size fraction would be inadvertently masked and lead to misleading outcomes in that size fraction.

Reply: Thank you for this very important point. We agree that the non-recovery of viral RNA from particles $<1 \mu\text{m}$ could be due to the reduced efficiency of extracting viruses from filters as compared to extracting viruses adhered to the wall of the 1.5 mL and 15 mL centrifuge tubes, where particles 1-4 μm and $>4 \mu\text{m}$ in diameter are captured using the NIOSH aerosol sampler. Additionally, to our knowledge, this is the first time the NIOSH samplers have been used to capture coronaviruses. Therefore, no baseline data exist for airborne coronavirus sampling using these samplers, limiting our understanding of the negative results in the $<1 \mu\text{m}$ size fraction. The above explanation has been added to the discussion section (lines 197-204).

Line 84-88: should refer to the SI directly here because it is difficult to interpret the sampling setup without seeing Figure S1 (or it seems that there is no Figure in the SI that corresponds to Patient 1 but only patients 2 and 3? This is confusing to me).

Reply: The set-up in Figure S1 depicts the layout of the room, which was constant throughout all air sampling cycles. The caption has been further clarified to specify that patient 1 was seated in the chair, while patients 2 and 3 were lying in bed during air sampling. Line has been added at the start of air sampling methods to refer to Figure S1 (line 82).

Figure s1: Also in Figure S1 it is unclear what “A” means in red circles. “A” = “air sampler” perhaps? Please clarify in the caption.

Reply: Figure S1 has been edited to include positioning of SKC samplers. Red circles have been re-labelled “N” for NIOSH air samplers and “S” for SKC air samplers.

Line 88-91: I didn't see the location of these other non-size-segregated SKC 37 mm PTFE filter cassettes in the SI diagram of sampling. Please add and generally improve the visual description of the sampling campaign to help best interpret findings. (I also never saw results from these samples presented anywhere either).

Reply: Figure S1 has been edited to include the non-size-segregated SKC samplers (with 37mm PTFE filter cassettes). The results from these samples have also been included in the results section (lines 150-151) and Table 1.

Line 98: pooling of the 6 NIOSH samples... why was this done? It seems a missed opportunity to get some vertical variations. Were you worried about recovering enough viral copies to detect?

Reply: Yes, as the air in the patient rooms was highly diluted, pooling of the NIOSH samples was carried out to improve recovery of an adequate number of viral copies for detection. Our approach was also limited by the number of available samplers, but we hope to continue refining our method such that we can explore other important questions regarding the spatial distribution of aerosols in this environment.

Line 101-102: what area of surface was swabbed and how does it compare to typical guidance? For example, WHO recommends sampling 25 cm² (https://apps.who.int/iris/bitstream/handle/10665/331058/WHO-2019-nCoV-Environment_protocol-2020.1-eng.pdf?sequence=1&isAllowed=y). Was at least a consistent amount of area swabbed? This could have important implications for the finding in Figure S3 that clinical and environmental Ct values were not correlated over time (i.e., if different surfaces were swabbed with different areas then that could be masking some association, which would just mean that clear conclusions can't necessarily be drawn).

Reply: The area of surface swabbed varied depending on site sampled as it is not feasible to standardize surface area for different sampling sites (e.g. switches and stethoscopes cannot have the sample surface area as floor or table, due to the nature and shape of the objects). However, a consistent sampling method was used, by the same study investigators, across different patients and rooms for each sampling site, minimizing inter-room variability. The area of the floor sampled was kept constant at 30cm².

Line 134-135: this specific description of environmental sampling in 3 AIIRs in the ICU and 27 AIIRs in the general ward, and also air sampling in 3 of the 27 AIIRs in the general ward, is very useful language and I suggest it be used in the abstract to increase specificity in abstract readers of the description of where samples were taken and where virus was recovered.

Reply: This specific description has been added to the abstract (lines 37-38, line 43).

Lines 141-142: this is where concentration estimates could vary potentially if viral recovery from the $<1\mu\text{m}$ filter was low, although I was surprised to not see any data reported from the $<1\mu\text{m}$ size range in Table 1. These results are surprisingly just not shown, which is a big omission in my opinion. Why weren't they shown?

Reply: Apologies for the oversight. All samples in the $<1\mu\text{m}$ range were negative, and we have updated Table 1 to include these results.

Line 149-153 and Figure 1: no quantification of concentrations was conducted for environmental contamination? (i.e., copies per cm^2) Why not?

Reply: No quantification of concentrations was done due to the variance in surface area swabbed between sampling sites (also mentioned above).

REVIEWER #2:

This paper includes important information on virus RNA detections in the air and surfaces in patient rooms in Singapore. The sample size of 30 patients is larger than some previous studies, but a greater number of measurements particularly _longitudinal_ measurements would have strengthened the study. There are a number of limitations to the conclusions

Reply: We agree that longitudinal measurements would have further strengthened the study, and cited this as a limitation in the discussion (lines 228-230).

1. Virus was not cultured and therefore infectivity cannot be confirmed. While air samples are known to be difficult to culture (as noted in lines 195-7) that would not apply to the surface samples?

Reply: Thank you for pointing this out. We agree that surface samples are more easily cultured as they are not subjected to the environmental stressors and shear forces involved in air sample collection (using air pumps which apply negative pressure). However, due to biosafety regulations and limited funding, we did not have immediate approval or support to isolate live virus from any of our samples. As we understand the importance of determining infectivity, follow-on studies have been designed and funding is being sought to isolate virus from both air and surface samples (line 226).

2. Was the greatest frequency of surface contamination actually on the non-touch surfaces (Figure 1), suggesting that even on the high-touch surfaces the presence of virus could have been through deposition of droplets rather than by contaminated touches? The lack of surface contamination in the highly-ventilated ICU rooms further supports this hypothesis?

Reply: We agree that the presence of virus on high-touch surfaces could be through deposition of droplets, in addition to contaminated touches. This point has been expanded upon in the discussion (lines 205-210).

3. Figure 3 would be easier to interpret if longitudinal data were available.

Reply: Unfortunately, longitudinal data were not collected as this was a cross-sectional study and we did not track environmental contamination over the course of illness for individual patients. This has been cited as a limitation in the discussion (lines 228-230).

REVIEWER #3:

In this manuscript the authors describe a study of 30 AIIR rooms occupied by COVID-19 patients and tested for SARS-CoV-2 to identify risk factors for environmental contamination by the virus. 245 surface samples from 27 rooms and aerosols samples in 3 rooms were examined showing 57% of rooms having at least 1 positive environmental sample and 2 of 3 rooms with positive airborne samples with particle in the 1-4 and >4 um size range. The data shows surface contamination is more prevalent in the first week of symptoms and the positive aerosol samples corroborates potential airborne transmission of the disease. The conclusions while not novel (multiple papers have been published) provides important convincing evidence and data that is needed by the HealthCare community in response to this rapidly moving respiratory disease.

1) The paper only presents PCR data and would be greatly enhanced by attempts to culture the virus to show infectivity.

Reply: We agree that this is a significant limitation of this study, and this has been cited in the discussion (lines 219-221). Attempts to isolate virus in viral culture from these samples are underway.

2) More description of the RT-PCR methods is needed, RT-PCR targeted envelope genes and and in house orf lab assay were used. Are they validated assays, how was the q-PCR done, e.g. the aerosol data is presented as the number of RNA copies/m³ with no information regarding what reference material is used for the qPCR.

Reply: The assay targeting the E gene is one of the available assays recommended by WHO for COVID-19 testing (Corman et al., 2020). The assay has been well validated with clinical specimens, and cross-checked with other coronaviruses and respiratory viruses, as reported in Corman et al., 2020. The in-house assay targeting the orf1ab was also validated with clinical specimens (Young et al., 2020) and cross-checked in a similar method. This in-house assay was recently approved by the Singapore Health Sciences Authority for in vitro diagnostic use (provisional license MDPA 2020-04). The description of laboratory methods has been expanded in the methods section (lines 123-128).

3) While the statistics seem appropriate, the description is limited. Figure 1 and 2 do not have an indication for variability only significance.

Reply: Figure 1 has no test of statistical significance and was used to demonstrate the areas of environmental contamination in rooms with any extent of environmental contamination.

Figure 2 was derived using overall proportion for each of the groups and χ^2 test was used, thus there is no 95% confidence interval depicted.

4) For the surface samples some indication of the amount of area sampled should be provided. Was a standard template used or just random swabbing of the surface.

Reply: As mentioned above, the area of surface swabbed varied depending on site sampled as it is not feasible to standardize surface area for different sampling sites. However, a consistent sampling method was used, by the same study investigators, across different patients and rooms for each sampling site, minimizing inter-room variability. The area of the floor sampled was kept constant at 30cm².

5) The data in table 2 suggests an inverse relationship between environmental contamination and O2 use, yet in line 156 the authors claim environmental contamination was not associated with the presence of symptoms or supplementary oxygen and in line 213 use the lack of association to conclude that in spite of O2 supplementation patients can still shed virus aerosols. This discrepancy should be cleared up.

Reply: This discrepancy has been duly noted, and the respective lines mentioning supplementary oxygen have been removed from the results (line 168) and the discussion (lines 240-241).

6) Line 149-150 should indicate that the air exhaust vent was also a location of significant contamination.

Reply: This has been included as suggested (line 161).

7) I think the fact that many of the contact surfaces that have been tested are in close proximity to the patients and could be contaminated by large droplets from patient coughs and sneezes and are not necessarily contaminated by touch needs to be mentioned in the discussion, the data could be demonstrating droplet spray and aerosol spread but the emphasis on "high touch" surfaces detracts from this concept.

Reply: We agree that the contamination of numerous contact surfaces could be via respiratory droplets. This point has been expanded upon in the discussion (lines 205-210).

8) minor typos. line 308 in in and line 315 95% not 955

Reply: These typographical errors have been corrected.

We look forward to your favourable review of our revised manuscript.

Thank you.

Yours Sincerely,

Dr Kalisvar Marimuthu

National Centre of Infectious Diseases, Singapore